# Mepirapim, a Novel Synthetic Cannabinoid, Induces Addiction-Related Behaviors through Neurochemical Maladaptation in the Brain of Rodents

**DOI:** 10.3390/ph15060710

**Published:** 2022-06-03

**Authors:** Kwang-Hyun Hur, YouYoung Lee, Audrey Lynn Donio, Shi-Xun Ma, Bo-Ram Lee, Seon-Kyung Kim, Jae-Gyeong Lee, Young-Jung Kim, MinJeong Kim, SeolMin Yoon, SooYeun Lee, Yong-Sup Lee, Seok-Yong Lee, Choon-Gon Jang

**Affiliations:** 1Department of Pharmacology, School of Pharmacy, Sungkyunkwan University, Suwon 16419, Korea; khh508@naver.com (K.-H.H.); luy902@naver.com (Y.L.); audreydonio@gmail.com (A.L.D.); shixun625@jhmi.edu (S.-X.M.); gift1201@daum.net (B.-R.L.); seonkyung_@naver.com (S.-K.K.); dksk179@naver.com (J.-G.L.); dhkcnghf@naver.com (Y.-J.K.); sylee@skku.edu (S.-Y.L.); 2School of Pharmacy, Keimyung University, 1095 Dalgubeoldae-ro, Dalseo-gu, Daegu 42601, Korea; alswjd91vov@naver.com (M.K.); sylee21@kmu.ac.kr (S.L.); 3Department of Fundamental Pharmaceutical Sciences, School of Pharmacy, Kyung Hee University, Seoul 02447, Korea; tjfalsdbs@naver.com; 4Department of Pharmacy, School of Pharmacy, Kyung Hee University, Seoul 02447, Korea; kyslee@khu.ac.kr; 5Department of Life and Nanopharmaceutical Sciences, School of Pharmacy, Kyung Hee University, Seoul 02447, Korea

**Keywords:** addiction, dopamine, γ-aminobutyric acid, synthetic cannabinoid, Mepirapim

## Abstract

Mepirapim is a synthetic cannabinoid that has recently been abused for recreational purposes. Although serious side effects have been reported from users, the dangerous pharmacological effects of Mepirapim have not been scientifically demonstrated. In this study, we investigated the addictive potential of Mepirapim through an intravenous self-administration test and a conditioned place preference test in rodents. Moreover, to determine whether the pharmacological effects of Mepirapim are mediated by cannabinoid receptors, we investigated whether Mepirapim treatment induces cannabinoid tetrad symptoms in mice. Lastly, to identify Mepirapim induced neurochemical maladaptation in the brains of mice, we performed microdialysis, western blots and neurotransmitter enzyme-linked immunosorbent assays. In the results, Mepirapim supported the maintenance of intravenous self-administration and the development of conditioned place preference. As a molecular mechanism of Mepirapim addiction, we identified a decrease in GABAeric signalling and an increase in dopaminergic signalling in the brain reward circuit. Finally, by confirming the Mepirapim-induced expression of cannabinoid tetrad symptoms, we confirmed that Mepirapim acts pharmacologically through cannabinoid receptor one. Taken together, we found that Mepirapim induces addiction-related behaviours through neurochemical maladaptation in the brain. On the basis of these findings, we propose the strict regulation of recreational abuse of Mepirapim.

## 1. Introduction

Cannabis, also known as marijuana, is the most popular recreational drug, used by over 200 million people [1]. Cannabis induces pleasurable feelings such as euphoria, excitement and relaxation [2]. These psychoactive effects are mainly attributed to the main component of cannabis, ∆-9-tetrahydrocannabinol (THC) [3]. Various synthetic cannabinoids (SCBs) have been developed to mimic the effects of THC. They are marketed as ‘Spices’ or ‘K2’ and have become one of the most commonly abused psychotropic drugs [4]. Although SCBs are often advertised as safe and legal, they have far more potent pharmacological effects than THC. Furthermore, their adverse effects are difficult to predict [5]. Indeed, there have been many reports of psychological side effects such as anxiety, depression and paranoia, as well as physical side effects such as tachycardia, hypertension, seizures and even death in SCB users [6,7].

SCB addiction has become a global public health problem as many people use SCB as a recreational drug [8]. SCB addicts frequently report withdrawal symptoms, including agitation, anxiety and irritability [9]. These adverse effects impair their quality of life and, paradoxically, make it difficult for them to limit drug use. The addictive potential of SCB is mainly attributed to its affinity for cannabinoid receptor type 1 (CB1R), which is abundantly expressed throughout the mesocorticolimbic system, known as the brain’s reward system [10]. The binding of SCB to CB1R suppresses the release of the inhibitory neurotransmitter, γ-aminobutyric acid (GABA) into dopaminergic neurons in the ventral tegmental area (VTA) [11]. This reduction in GABAergic inhibition activates dopamine (DA) signalling into the nucleus accumbens (NAc), a key region in the brain reward circuitry, leading to addictive effects [12]. Although these molecular mechanisms of SCB addiction have been clearly demonstrated, many SCBs remain erroneously advertised as non-addictive and are, therefore, abused for recreational purposes [13].

Mepirapim is a new-generation SCB first detected in illegal herbal mixtures in Japan, 2013 [14]. Mepirapim was classified as an SCB because of its affinity for the CB1R but was not considered to have addictive or dangerous effects because of this affinity being relatively low [15]. Nevertheless, this drug has recently been abused for recreational purposes [16]. Even more troubling, serious side effects have been reported from users, including circulatory failure, organ congestion, gastrointestinal bleeding and even death [17]. Considering its structural similarity with JWH-018 (Figure 1), which is a controlled Schedule 1 substance in the United States [18], Mepirapim has a high potential to exhibit psychotropic effects and dangerous side effects. Scientific investigation into the addictive potential of Mepirapim is necessary to prevent the misuse of this drug and to legally regulate its use.

In this study, we first investigated that Mepirapim induces addiction through reinforcing and rewarding effects. Second, we confirmed whether Mepirapim causes pharmacological effects via CB-receptor-mediated action. Third, we suggested a molecular mechanism for the abnormal behaviours induced by Mepirapim treatment, by confirming the neurochemical maladaptation in the brain of mice used in the behavioural experiments. Through this series of studies, we have demonstrated that Mepirapim has addictive potential, suggesting the need for the strict legal regulation of recreational abuse of Mepirapim.

## 2. Results

### 2.1. Mepirapim Treatment Supported Maintenance of IVSA in Rats

The average number of infusions was significantly higher in the Mepirapim groups (0.003, 0.01 and 0.03 mg·kg^−1^·inf^−1^) compared with the vehicle group (Figure 2B, F _(3,16)_ = 5.57, *p* < 0.05). Two-way ANOVA results revealed statistically significant effects of drug treatment (F _(3,16)_ = 5.57, *p* < 0.05) and day (F _(6,96)_ = 17.35, *p* < 0.05), but there was no significant effect of their interaction (F _(18,96)_ = 1.31, *p* = 0.2). Similarly, the average number of active lever presses was significantly higher in the Mepirapim groups (0.003, 0.01 and 0.03 mg·kg^−1^·inf^−1^) compared with the vehicle group (Figure 2C, F _(3,16)_ = 4.05, *p* < 0.05). Two-way ANOVA revealed statistically significant effects of drug treatment (F _(3,16)_ = 4.05, *p* < 0.05) and day (F _(6,96)_ = 22.6, *p* < 0.05), but there was no significant effect of their interaction (F _(18,96)_ = 1.01, *p* = 0.46). These results indicate that Mepirapim is an addictive drug with reinforcing effects [19].

Furthermore, the average number of inactive lever presses was significantly higher in the Mepirapim groups (0.01 and 0.03 mg·kg^−1^·inf^−1^) than in the vehicle group (Figure 2D, F _(3,16)_ = 3.8, *p* < 0.05), indicating that Mepirapim induces impulsive behaviours similar to those induced by treatment with other SCBs [20]. Two-way ANOVA revealed statistically significant effects from drug treatment (F _(3,16)_ = 3.8, *p* < 0.05) and day (F _(6,96)_ = 3.58, *p* < 0.05), but no significant effect of their interaction was observed (F _(18,96)_ = 0.81, *p* = 0.69).

### 2.2. Mepirapim Treatment Produced CPP in Mice

Mepirapim treatment at 0.3 and 1 mg·kg^−1^ increased CPP scores (sec) compared with those from vehicle treatment. More specifically, the Mepirapim group at 1 mg·kg^−1^ dose showed a significant difference from the vehicle group (Figure 3B, F _(3,32)_ = 6.21, *p* < 0.05). This result indicates that Mepirapim is an addictive drug with rewarding effects [21]. Conversely, the group treated with 3 mg·kg^−1^ Mepirapim showed a significantly reduced CPP score compared with the vehicle-treated group. This suggests that Mepirapim treatment at 3 mg·kg^−1^ may have induced aversive effects in mice.

### 2.3. Mepirapim Treatment Induced CB Tetrad-Related Symptoms in Mice

In OFT, the group treated with Mepirapim 3 mg·kg^−1^ dose showed a significantly reduced total distance moved (m) compared with the vehicle-treated group (Figure 3C, F _(3,32)_ = 16.39, *p* < 0.05). This indicates that Mepirapim treatment induces hypomotility. Additionally, treatment with 3 mg·kg^−1^ Mepirapim induced a significant decrease in time in the centre (%) compared with control, which indicates the anxiety-like behaviour of mice (Figure 3D, F _(3,32)_ = 5.5, *p* < 0.05). In body temperature measurements, Mepirapim treatment at 3 mg·kg^−1^ significantly reduced the body temperature (°C) compared with the vehicle treatment (Figure 3E). Two-way ANOVA revealed statistically significant effects of drug treatment (F _(3,32)_ = 27, *p* < 0.05), time (F _(7,224)_ = 11.39, *p* < 0.05) and their interaction (F _(21,224)_ = 3.7, *p* < 0.05). These results indicate that Mepirapim treatment at a dose of 3 mg·kg^−1^ induces CB tetrad symptoms, including hypomotility and hypothermia. Both of which can be aversive in mice.

### 2.4. Mepirapim Treatment Induced Changes in Extracellular Levels of DA and Its Metabolites (DOPAC and HVA) in Rat NAc

Mepirapim treatment increased DA levels in a dose-dependent manner. Treatment with 1 and 3 mg·kg^−1^ doses generated significant increases in AUC_DA_ compared with vehicle treatment (Figure 4B, F _(3,16)_ = 8.96, *p* < 0.05). Two-way ANOVA results revealed statistically significant effects of drug treatment (F _(1,8)_ = 22.15, *p* < 0.05), time (F _(9,72)_ = 4.76, *p* < 0.05) and their interaction (F _(9,72)_ = 2.55, *p* < 0.05). Accordingly, the Mepirapim group showed significantly higher DOPAC levels and AUC_DOPAC_ compared with the vehicle control group (Figure 4C, F _(3,16)_ = 7.87, *p* < 0.05). Two-way ANOVA revealed statistically significant effects of drug treatment (F _(1,8)_ = 18.95, *p* < 0.05), time (F _(9,72)_ = 2.29, *p* < 0.05) and their interaction (F _(9,72)_ = 2.51, *p* < 0.05). Similarly, Mepirapim treatment at 1 and 3 mg·kg^−1^ significantly increased HVA levels and AUC_HVA_ compared with the vehicle treatment (Figure 4D, F _(3,16)_ = 4.34, *p* < 0.05). Two-way ANOVA revealed statistically significant effects of drug treatment (F _(1,8)_ = 5.84, *p* < 0.05), time (F _(9,72)_ = 2.61, *p* < 0.05) and their interaction (F _(9,72)_ = 2.1, *p* < 0.05).

### 2.5. Mepirapim Treatment Induced Changes in Addiction-Related Molecules in Mice VTA and NAc

In these experiments, the molecular changes in the VTA and NAc of mice treated with vehicle or Mepirapim (1 and 3 mg·kg^−1^) during the CPP and tetrad tests were investigated. Western blot tests revealed that Mepirapim treatment induces significant maladaptation of proteins related to GABA and DA. In the VTA, Mepirapim treatment significantly increased expression levels of CB1R (Figure 5A, F _(2,12)_ = 10.29, *p* < 0.05) and TH (Figure 5C, F _(2,12)_ = 8.43, *p* < 0.05). Moreover, the same treatment caused a significant decrease in expression levels of GABA_A_ (Figure 5B, F _(2,12)_ = 20.96, *p* < 0.05) compared with vehicle treatment. In the NAc, the Mepirapim-treated groups showed significantly increased expression levels in CB1R (Figure 5D, F _(2,12)_ = 144.8, *p* < 0.05), GAD (Figure 5E, F _(2,12)_ = 9.7, *p* < 0.05) and D1DR (Figure 5F, F _(2,12)_ = 12.9, *p* < 0.05) compared with the vehicle-treated group.

Neurotransmitter ELISA results revealed that Mepirapim treatment induces an increase of both GABA (Figure 6A, F _(2,12)_ = 3.93, *p* < 0.05, Figure 6C, F _(2,12)_ = 8.11, *p* < 0.05) and DA (Figure 6B, F _(2,12)_ = 62.64, *p* < 0.05, Figure 6D, F _(2,12)_ = 16.41, *p* < 0.05) levels in the VTA and NAc, in a dose-dependent manner. More specifically, a significant difference was observed between the Mepirapim 3 mg·kg^−1^ treatment group and the vehicle treatment group.

## 3. Discussion

Recently, the recreational misuse of SCBs has become a global health problem, causing frequent emergencies [7]. Mepirapim has also been reported to cause serious side effects [17]. Nevertheless, this drug is still incorrectly believed to cause no dangerous pharmacological effects [15]. In this study, we have demonstrated the risk of Mepirapim abuse by confirming that this drug induces addiction-related behaviours in rodents.

From IVSA test, we found that Mepirapim supports the maintenance of IVSA with a high number of active lever presses, suggesting a potent reinforcing effect [19]. This is consistent with previous studies showing that addictive SCBs, such as JWH-018 and WIN 55,212-2, support IVSA [22,23]. In addition, Mepirapim treatment-induced impulsive behaviours identified by multiple inactive lever presses, which are also induced by other SCBs treatment [20]. These impulsive behaviours are frequently observed in SCB users and are considered to increase the risk of recreational abuse of the drug [24]. As a molecular mechanism for these pharmacological effects of Mepirapim, we confirmed that this drug increased the levels of DA and DA metabolites in the NAc, which is the core of the DAergic reward circuit [25]. This enhancement of DA signalling by SCB treatment has been confirmed in previous studies. Indeed, it is considered to be the most important molecular mechanism of SCB addiction [26,27], and Mepirapim is also thought to have the same mechanism of action.

From the CPP test, we found that Mepirapim has rewarding effects producing CPP, which was also confirmed in other SCBs [23,28]. Interestingly, only low doses of Mepirapim (0.3 and 1 mg·kg^−1^) induced CPP whereas higher doses (3 mg·kg^−1^) rather induced conditioned place aversion. This shift from preference to aversion according to the dose increase was also confirmed in other SCBs [29], and it is considered that the dose increase induces stronger aversive effects than the rewarding effects of the drug. As experimental results supporting this hypothesis, Mepirapim 3 mg/kg induced negative physical symptoms, including hypomotility and hypothermia, that were not induced by lower doses of Mepirapim. Taken together, these results indicate that Mepirapim induces rewarding effects at low doses and, conversely, aversive effects at high doses. Additionally, the result that Mepirapim induced CB tetrad symptoms indicates that Mepirapim strongly induces CB1R-mediated pharmacological effects. Notably, this is in distinct contrast to previous studies suggesting that Mepirapim has a weak affinity for CB receptors [15]. The action of Mepirapim on CB1R is further supported by the results that Mepirapim treatment increased the expression of CB1R in both the VTA and NAc. Additionally, Mepirapim is considered to inhibit the action of GABA through these CB1R-mediated effects. This is confirmed by the decreased expression of GABA_A_ required for GABA signalling in the VTA. Subsequently, because of homeostatic actions to normalize GABA function, it is assumed that the expression of GAD, the GABA synthesizing enzyme, was increased in NAc to increase overall GABA levels. This CB-induced decrease in GABA signalling with a simultaneous increase in GABA levels as a compensatory action was also confirmed in previous studies [30]. Consequently, this reduction of GABAergic inhibition by Mepirapim would lead to hyperactivity of DA signalling, which could be supported by increased D1DR in the NAc, increased TH, the DA synthesizing enzyme, in the VTA and increased DA levels in both regions. This hyperactivity of the DA system by Mepirapim is thought to be an important molecular mechanism of Mepirapim addiction, consistent with other addictive SCBs [23,31].

## 4. Materials and Methods

### 4.1. Animals

Experimental animals were selected on the basis of previous studies with consideration of the sensitivity to each behavioural experiment [32]. Studies were designed to generate equally sized groups using randomization. Male Sprague Dawley rats (7 weeks old, 250–270 g) were purchased from Orient Bio Co., Ltd. (Seoul, Korea). The rats were used for intravenous self-administration (IVSA) tests and microdialysis. The rats were housed individually in cages (26 × 42 × 18 cm) with free access to food and water, except during food training. Male C57BL/6J mice (7 weeks old, 20–22 g) were purchased from Dae Han Biolink Co., Ltd. (Eumseong, Korea), for conditioned place preference (CPP) tests and tetrad tests. Mice were housed with six mice per cage (26 × 42 × 18 cm) and allowed access to water and food *ad libitum.* Both rats and mice were maintained in a temperature- and humidity-controlled room (23 °C ± 1 °C, 55% ± 5%) under a 12-h light/dark cycle (lights on 07:00–19:00). Upon arrival, animals were acclimated for a week before the start of behavioural experiments. On the test day, animals were introduced to the test site for acclimatization 60 min before starting the test.

### 4.2. Drugs

Mepirapim hydrochloride (98.7% HPLC Purity) was synthesized and provided by Professor Yong-Sup Lee at the Medicinal Chemistry Laboratory, Department of Pharmacyand Department of Life and Nanopharmaceutical Sciences, College of Pharmacy, Kyung Hee University (Seoul, Korea). This drug was dissolved in physiological saline (0.9%).

### 4.3. Abuse Potential Assessments

#### 4.3.1. IVSA Test

An IVSA test was performed to assess the reinforcing effects of Mepirapim, according to a previously reported experimental design [33]. Figure 2A shows the detailed experimental procedure.

##### Apparatus

The IVSA test was conducted in a standard operant chamber, which is enclosed in a light- and sound-attenuating cubicle with ventilation fans (28 × 26 × 20  Med Associates Inc., St. Albans, VT, USA). Chambers consisted of front and back walls made of transparent plastic. The other walls were made of opaque metal. In each chamber, one of the side opaque walls contained two metal retractable levers (4.8 × 1.9 cm), which are active or inactive. A cue light (3 W, 28 V) was positioned above each lever, and a house light (3 W, 28 V) was located on the opposite wall. Drug injections were delivered by a syringe pump (Razel Scientific Instruments, Georgia, VT, USA) located on top of the cubicle. All IVSA sessions were controlled and recorded in the experimental room using a PC with a custom interface and software.

##### Food Training

Food training preceded surgery, as described in the previous study, to facilitate the acquisition of operant responding [33]. Rats were trained to press a lever to obtain 45 mg of food pellets (Bio-Serv, Frenchtown, NJ, USA). Before beginning training, rats were deprived of food for 12 h and then trained in standard operant chambers equipped with a food pellet dispenser until the desired criteria were satisfied (80 food pellets over three consecutive days) in 1-h daily sessions. Following food training, food pellet dispensers were removed from the chambers and rats were returned to ad libitum feeding conditions.

##### Intravenous Catheterization

Following the complete acquisition of food training, rats were anaesthetized with pentobarbital (50 mg/kg, i.p.; Hanlim Pharm Co., Ltd., Seoul, Korea). Under anaesthesia, rats were implanted with indwelling silastic catheters (0.3 mm ID × 0.64 mm OD; Dow Corning, Midland, TX, USA) into the left external jugular vein and secured with a surgical suture thread. The outer part of the catheter was subcutaneously exited to the rat’s back. Catheters were flushed daily with 0.2 mL of gentamicin sulfate (0.32 mg/mL; Shin Poong Pharm Co., Ltd., Seoul, Republic of Korea) in heparinized saline (30 IU/mL) to prevent occlusion and infection. Rats were allowed to recover at least 7 days before beginning drug self-administration.

##### IVSA Test for 7 Days

After recovery from surgery, animals (*n* = 5 per group) were randomly assigned into four groups for drug IVSA tests. A vehicle (saline, i.v.) group was used as a negative control. Three Mepirapim groups received various infusion concentrations (0.003, 0.01 and 0.03 mg·kg^−1^·infusion^−1^, i.v.). The IVSA sessions were conducted under a fixed ratio of 1 schedule of reinforcement for 2 h per day on seven consecutive days. Daily, each rat was placed in a standard operant chamber and the catheter was connected to Tygon tubing suspended from a balance arm above the chamber. Each session started by inserting the two levers and illuminating the house light. Pressing the right (active) lever paired with the cue light initiated intravenous infusion of drugs using a syringe pump. Each 0.1 mL of drug infusion lasted 4 s and was followed by a 16-s time-out period. After the time-out period, the cue light was turned off and the house light was turned on, signalling that the next infusion is available. Left (inactive) lever presses were recorded but had no programmed consequences. Sessions ended by retracting the two levers.

#### 4.3.2. CPP Test

A CPP test was performed to evaluate the rewarding effects of Mepirapim according to a previously reported experimental design [34]. The apparatus for the CPP test comprised two equally sized compartments (15 × 15 × 15 cm) separated by two removable guillotine doors. One compartment had a white background with a stainless steel mesh floor, whereas the other had a black background with a stainless steel grid floor. Both compartments were placed under conditions of dim illumination (12–13 lux). The experimental procedure is shown schematically in Figure 3A and comprised these three following phases: baseline test (day 1), conditioning (days 2 to 9) and preference test (day 10). All sessions were conducted in a sound isolated experimental room between 10 am and 6 pm. The data were analysed using a computer-based video tracking system (NeuroVision, Pusan National University, Busan, Korea). On day 1, mice were allowed to move freely between both compartments for 20 min. The time spent in each compartment during this session was recorded to establish the baseline preferences. Animals were then randomly assigned to five groups (*n* = 9 per group) in an unbiased counterbalanced design. A vehicle (saline, i.p.) group served as a negative control. Then, three distinct Mepirapim (0.3, 1 and 3 mg·kg^−1^, i.p.) groups received various dosages of the drug. On days 2–9, conditioning (45 min per session) was performed with the guillotine door being closed. During this period, the Mepirapim groups were injected with either Mepirapim or vehicle on alternate days. The vehicle group was injected with the vehicle every day. All injections were administered immediately before being placed in the paired compartment. On day 10, a preference test was performed by allowing mice free access to both compartments for 20 min. Here again, the time spent in each compartment during the 20 min session was measured to establish postconditioning preferences. The CPP score was defined by subtracting the time spent on the drug-paired compartment during the preference test session from the time spent on the same compartment during the baseline test session.

### 4.4. Assessment of CB Tetrad-Related Symptoms

The CB tetrad is a series of behavioural symptoms consisting of hypomotility, hypothermia, catalepsy and analgesia [35]. To determine whether Mepirapim exerts CB receptor-mediated effects, we investigated whether administration of Mepirapim induces hypomotility and hypothermia among tetrad symptoms. Tetrad tests were conducted after the CPP test, and both mice and the experimental group were used the same (*n* = 9 per group): a vehicle (saline) group served as a negative control and three Mepirapim (0.3, 1 and 3 mg·kg^−1^) groups. Mice were sacrificed after the completion of the behavioural tests for further molecular studies.

#### 4.4.1. Open Field Test

The open field test (OFT) was performed to determine whether Mepirapim treatment induces hypomotility. The test used previously reported experimental designs, with minor modifications [36]. The open field was an opaque plastic box (30 × 30 × 30 cm), divided into 16 (4 × 4) equal sectors (7.5 × 7.5 cm). The field was subdivided into central and peripheral sectors. The central sector contained four central squares (2 × 2), and the peripheral sector contained the remaining squares. The experiment was conducted under dim lighting (12–13 lux) conditions. Immediately after drug treatment, the mice were placed in the corner of the open field and allowed to explore freely for 60 min. Their movements were recorded and analysed using a video tracking system (NeuroVision, Pusan, Korea). General locomotor activity was evaluated by the total distance moved over 60 min. Anxiety-like behaviour was assessed by quantifying the time in the centre (%) during the first 10 min. Time spent in the centre (%) was calculated by dividing the time spent in the centre sector by the time spent in all sectors. Open fields were thoroughly cleaned with 70% ethanol between tests.

#### 4.4.2. Body Temperature Measurement

Body temperature (°C) measurements were taken, with minor modifications of previously reported experimental designs, to determine whether Mepirapim treatment induces hypothermia [36]. After the basal body temperature was measured, mice were treated with the drug. The temperature was then measured for 150 min afterwards, at 15 min intervals. The experimenter held the animal loosely and inserted a lubricated rectal probe (World Precision Instruments Inc., Sarasota, FL, USA) into the animal’s rectum to a depth of 2 cm. The temperature was measured with a thermometer (Physitemp Instruments, LLC., Clifton, NJ, USA) and recorded when the fluctuation stopped.

### 4.5. Microdialysis

This test was performed to determine whether Mepirapim treatment changed extracellular levels of the neurotransmitters and their metabolites in rat NAc (*n* = 5 per group). The test was performed with minor modifications to a previously reported experimental design [37].

#### 4.5.1. Chemicals

Dopamine (DA), 3,4-dihydroxyphenylacetic acid (DOPAC), homovanillic acid (HVA), DA-*d*_4_, DOPAC-*d*_5_, HVA-*d*_5_ and ascorbic acid were purchased from Sigma–Aldrich. Acetonitrile and methanol (HPLC grade) were purchased from Fisher Scientific (Waltham, MA, USA) and Merck (Darmstadt, Germany), respectively. All other chemicals and solvents were of reagent grade.

#### 4.5.2. Probe Implantation

For surgery, the rats were anaesthetized by pentobarbital anaesthesia (50 mg·kg^−1^, i.p.), and a microdialysis probe guide cannula (CMA 11; CMA Microdialysis AB, Kista, Sweden) was stereotaxically implanted into the brain. After a six-day recovery period following the surgery, a microdialysis probe (membrane length, 2 mm; cutoff, 6 kDa; CMA Microdialysis AB) was inserted into the NAc shell (AP + 1.7 mm, ML + 0.8 mm, from the bregma; DV − 6.0 mm, from the skull) through the guide cannula of anaesthetized rats.

#### 4.5.3. Brain Microdialysis

Artificial cerebral spinal fluid (a mixture of 150 mM sodium chloride, 3.0 mM potassium chloride, 1.4 mM calcium chloride and 0.8 mM magnesium chloride in 10 mM phosphate buffer (pH 7.4)) was perfused at 1.5 μL·min^−1^ using a microinjection pump (CMA 100; CMA Microdialysis AB) for 2 h for stabilization. Six baseline samples were collected every 20 min for 2 h. Subsequently, Mepirapim was administered every hour with a gradually increasing dose (0.3, 1 and 3 mg·kg^−1^, i.p.). Microdialysate was collected at 20 min intervals (Figure 4A). At the termination of experiments, all rats were sacrificed for histological confirmation of microdialysis probe location.

#### 4.5.4. LC-MS/MS Analysis

A total of 25 μL of the microdialysates collected from rats were mixed with 5 μL of the internal standard solution (a mixture solution of deuterated compounds) and analysed by a fully validated liquid chromatography (LC)−tandem mass spectrometry (MS/MS), using a 1260 infinity LC system and 6460 triple-quadrupole MS/MS (Agilent Technologies, Santa Clara, CA, USA) coupled with a 1260 infinity extra binary pump and degasser (Agilent Technologies). The XBridge BEH HILIC Sentry Guard Cartridge 130 Å (4.6 × 20 mm, 3.5 μm; Waters, Milford, MA, USA) and the Atlantis T3 column (2.1 × 100 mm, 3 μm; Waters) were applied as sample enrichment and separation columns, respectively. The mobile phases (A, 5 mM ammonium formate/0.1% formic acid in water; B, 0.1% formic acid in acetonitrile) were passed through both the enrichment and separation columns with the following gradient conditions: 0–1.0 min, 5% B; 1.0–6.5 min, 5–90% B; 6.5–7.5 min, 90% B; 7.5–7.6 min, 90–5% B; 7.6–11.5 min, 5% B. The MS/MS system was operated using electrospray ionization in the polarity-switching mode (DA, positive; DOPAC and HVA, negative). The MS/MS conditions were optimized as follows: drying gas temperature, 350 °C; drying gas flow, 10 L·min^−1^; nebulization pressure, 35 psi; capillary voltage, 4.5 kV; temperature of sheath gas, 250 °C; and sheath gas flow, 5 L·min^−1^. Multiple reaction monitoring was used for quantification. Each analytical stock solution (1 mg·mL^−1^) was prepared in 1 mM ascorbic acid in a 1:1 solution of water and methanol to prevent oxidation. Solutions were stored at −80 °C before analysis.

The LC−MS/MS data for the measurements of DA, DOPAC and HVA were processed using MassHunter software (B.04.00, Agilent Technologies). Baseline values were determined from three consecutive microdialysates in which the concentration of neurotransmitters fluctuates below 20% during the stabilization session. Neurotransmitter levels (%) in each sample were quantified as a percentage of the baseline value. The area under the curve (AUC) for each neurotransmitter was calculated for each period (AUC_0–60_, AUC_60–120_ and AUC_120–180_), accompanied by changes in drug concentration (0.3, 1 and 3 mg·kg^−1^). For comparison, the AUC of the vehicle group was calculated as the AUC_0–180_ of the entire period divided into thirds.

### 4.6. Western Blot

This test was performed to determine whether Mepirapim treatment changed protein expression in different brain regions (VTA and NAc). The procedure was slightly modified from previously reported experimental designs [34]. Brain samples from mice treated with either the vehicle or Mepirapim (1 or 3 mg·kg^−1^) during the CPP and tetrad tests were used (*n* = 5 per group). Protein concentrations were measured using a protein assay kit (Thermo Scientific, Rockford, IL, USA). Samples containing 5 μg of protein were separated in 8–10% SDS–polyacrylamide gels and transferred to polyvinylidene difluoride transfer membranes in transfer buffer (25 mM Tris−HCl buffer containing 192 mM glycine and 20% *v*/*v* methanol) at 4 °C for 1 h. Membranes were blocked with 5% nonfat milk containing 0.5 mM Tris−HCl (pH 7.5), 150 mM NaCl and 0.1% Tween-20 and incubated at room temperature (20 °C–25 °C) for 1 h. The membrane was subsequently incubated with primary antibodies to β-actin (1:1000, Santa Cruz Biotechnology, Dallas, TX, USA, Cat# sc-47778, RRID: AB_626632), cannabinoid receptor 1 (CB1R) (1:1000; Abcam, Cambridge, UK, Cat# ab23703), RRID: AB_447623), dopamine receptor D1 (D1DR) (1:1000; Abcam, Cambridge, UK, Cat# ab216644), γ-aminobutyric acid receptor A (GABA_A_) (1:1000; Abcam, Cambridge, UK, Cat# ab154822), RRID: AB_447623), glutamate decarboxylase (GAD) (1:5000, Millipore, Burlington, Middlesex County, MA, USA, Cat# MAB5406, RRID: AB_2278725), or tyrosine hydroxylase (TH) (1:2000, Cell Signalling Technology, Danvers, MA, USA, Cat# 2792, RRID:AB_2303165) on a shaker (100 rpm) in a refrigerator (2 °C–8 °C) for 15 h. After five washes with Tris-buffered saline with 0.1% Tween-20 (TBST), membranes were incubated with goat anti-rabbit (1:3000, Cell Signalling Technology) or anti-mouse (1:10,000, GenDEPOT, Katy, TX, USA) horseradish peroxidase (HRP)-conjugated secondary antibodies in TBST with 5% non-fat milk at room temperature (20 °C–25 °C) for 1 h. Membranes were washed again five times in TBST buffer. Bands were visualized using enhanced chemiluminescence (ECL) detection by exposure to a mixture of ECL reagents A and B (Anigen, Hwaseong, Korea) at a 1:1 ratio, followed by contact with photographic film (Kodak, Rochester, NY, USA) for a few minutes. Protein bands were quantified with densitometric analysis using Image J software from NIH (Bethesda, MD, USA).

### 4.7. Neurotransmitter Enzyme-Linked Immunosorbent Assay

This test was performed to determine whether Mepirapim treatment changed neurotransmitter levels in each brain region (VTA and NAc). Minor modifications were made to the previously reported experimental design [32]. Brain samples from mice treated with the vehicle or Mepirapim (1 or 3 mg·kg^−1^) during the CPP and tetrad tests were used (*n* = 5 per group).

#### 4.7.1. GABA Enzyme-Linked Immunosorbent Assay

GABA levels were measured quantitatively using an enzyme-linked immunosorbent assay (ELISA) kit (ImmuSmol, Talence, France, Cat # BA E-2500). This test was performed according to the manufacturer’s instructions. The absorbance was read with an ELISA reader (SpectraMax M2, Molecular Devices, San Jose, CA, USA) at 450 nm. GABA levels in the samples were quantified by comparing the absorbance to a reference curve prepared with standard concentrations.

#### 4.7.2. DA ELISA

DA levels were measured quantitatively using an ELISA kit (ImmuSmol, Talence, France, Cat # BA E-5300). This test was performed according to the manufacturer’s instructions. The absorbance was read with an ELISA reader (SpectraMax M2, Molecular Devices, San Jose, CA, USA) at 450 nm. GABA levels in the samples were quantified by comparing the absorbance to a reference curve prepared with standard concentrations.

### 4.8. Data and Statistical Analysis

The data and statistical analyses comply with the recommendations on experimental design and analysis in pharmacology [38]. The declared group size is the number of independent values. Statistical analysis was done using these independent values. All analyses were performed only for studies with *n* = 5 or more in each group size, using Prism 6.0 software (GraphPad Software, Inc., San Diego, CA, USA) by researchers blind to the origin of data. Statistical analyses were performed using analysis of variance (ANOVA), and the statistical significance was set to *p* < 0.05. When F achieved a *p* < 0.05 and there was no significant variance in homogeneity, ANOVA was followed by Fisher’s Least Significant Difference (LSD) post hoc test.

## 5. Conclusions

In this study, we confirmed that Mepirapim induces addiction by causing an imbalance between the GABA system and the DA system through its potent actions on CB1R. Based on these scientific demonstrations of Mepirapim addiction, we suggest the strict regulation of recreational abuse of SCBs, including Mepirapim. Furthermore, the results of this study should be widely publicized so that substance abusers can recognize for themselves that novel psychotropic drugs have dangerous pharmacological effects, including addictive properties.

## Figures and Tables

**Figure 1 pharmaceuticals-15-00710-f001:**
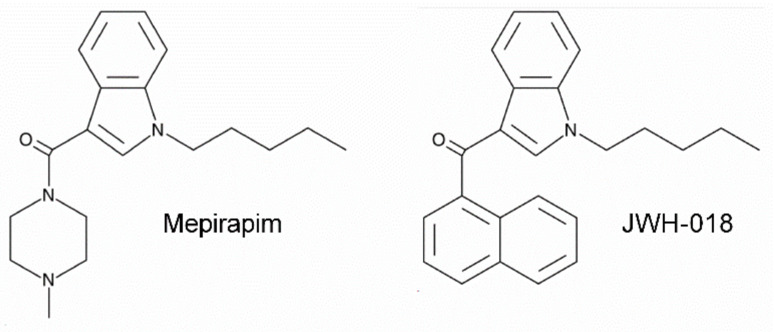
Chemical structures of Mepirapim and JWH-018.

**Figure 2 pharmaceuticals-15-00710-f002:**
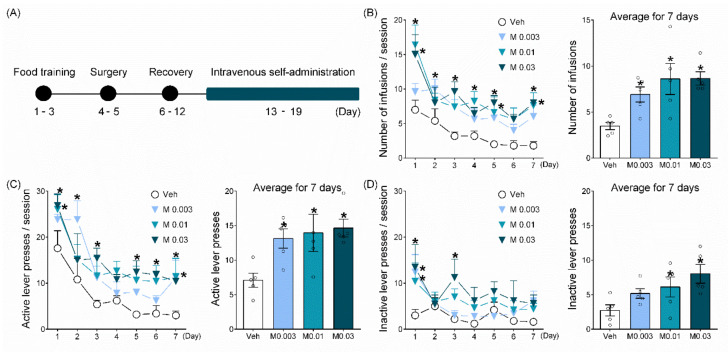
**Mepirapim treatment supported the maintenance of IVSA in rats.** (**A**) Experimental schedule. Male Sprague Dawley rats (7 weeks old, 250–270 g, *n* = 5 per group) were used. Each group of rats self-administered either vehicle or Mepirapim (0.003, 0.01 and 0.03 mg·kg^−1^·infusion^−1^, i.v.) under a fixed ratio 1 schedule for seven consecutive days. All drugs were injected intravenously at a volume of 0.1 mL·infusion^−1^. (**B**) Number of infusions during each session and average for 7 days. (**C**) Number of active lever presses during each session and average for 7 days. (**D**) Number of inactive lever presses during each session and average for 7 days. Data are presented as means ± SEMs. Significant differences between the vehicle group and other groups are indicated by * *p* < 0.05.

**Figure 3 pharmaceuticals-15-00710-f003:**
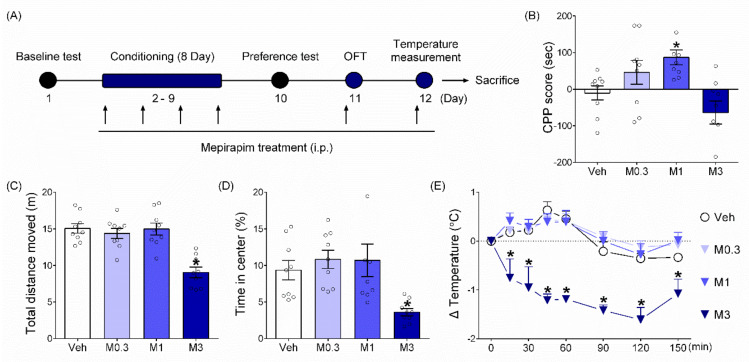
**Mepirapim treatment induced the development of CPP and tetrad symptoms in mice.** (**A**) Experimental schedule. Each arrow represents a drug treatment. Male C57BL/6J mice (7 weeks old, 20–22 g, *n* = 9 per group) were used. (**B**) CPP test. Mice were conditioned with either vehicle or Mepirapim (0.3, 1 and 3 mg·kg^−1^, i.p.) during the conditioning period. (**C**,**D**) Open field test (OFT). Immediately before the test, mice were treated with vehicle or Mepirapim. (**E**) Body temperature measurement. After basal body temperature was measured, mice were treated with vehicle or Mepirapim. Data are presented as means ± SEMs. Significant differences between the vehicle group and other groups are indicated by * *p* < 0.05.

**Figure 4 pharmaceuticals-15-00710-f004:**
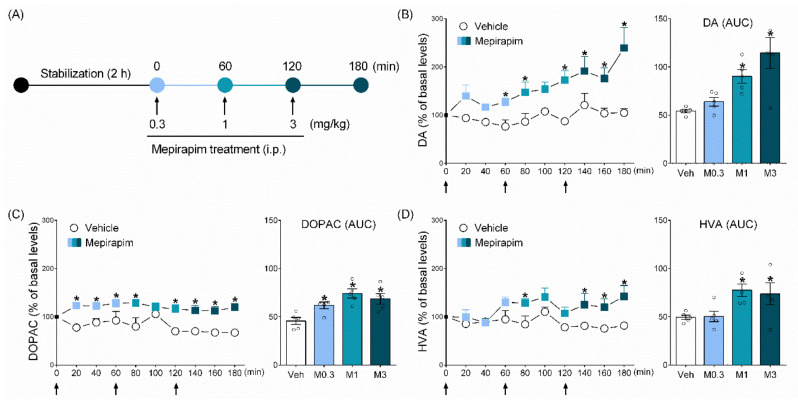
**Mepirapim treatment-induced changes in levels of neurotransmitters in rat NAc.** (**A**) Experimental schedule for microdialysis in the rat NAc. After baseline samples were collected, rats were treated with progressively increasing doses of Mepirapim (0.3, 1 and 3 mg·kg^−1^, i.p.) every hour (*n* = 5 per group). Each arrow represents a drug treatment. Concentration–time profiles and areas under the concentration–time curves (AUC) for (**B**) dopamine (DA), (**C**) 3,4-dihydroxyphenylacetic acid (DOPAC) and (**D**) homovanillic acid (HVA). Data are presented as means ± SEMs. Significant differences between the vehicle group and other groups are indicated by * *p* < 0.05.

**Figure 5 pharmaceuticals-15-00710-f005:**
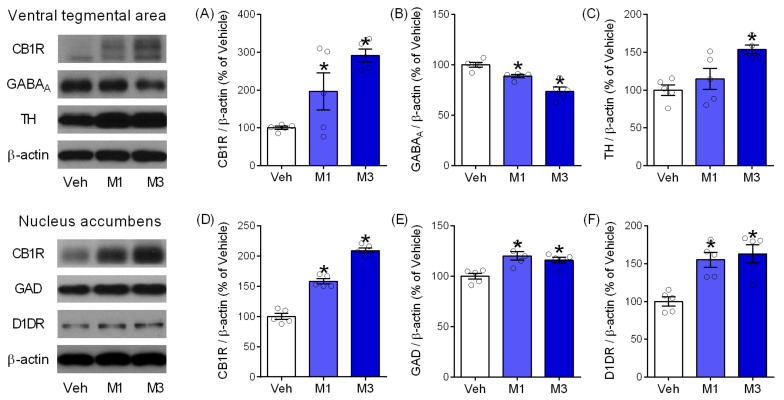
**Mepirapim treatment-induced changes in the expression of proteins related to DA and GABA in mouse VTA and NAc.** Mice in each group were treated with either vehicle or Mepirapim (1 and 3 mg·kg^−1^, i.p.) during the CPP test and tetrad test, respectively (*n* = 5 per group). Western blot analysis of protein expression in VTA (**A**–**C**) and NAc (**D**–**F**). Data are presented as means ± SEMs. Significant differences between the vehicle group and other groups are indicated by * *p* < 0.05. CB1R: cannabinoid receptor 1, D1DR: dopamine receptor D1, GABA_A_: γ-aminobutyric acid receptor A, GAD: glutamate decarboxylase, TH: tyrosine hydroxylase.

**Figure 6 pharmaceuticals-15-00710-f006:**
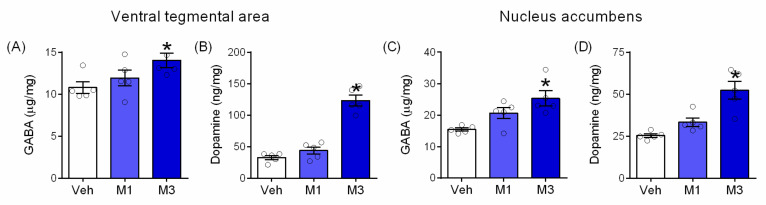
**Mepirapim treatment-induced changes in DA and GABA levels in mouse VTA and NAc.** Mice in each group were treated with either vehicle or Mepirapim (1 and 3 mg·kg^−1^, i.p.) during the CPP test and tetrad test, respectively (*n* = 5 per group). Measurement of neurotransmitter levels in VTA (**A**,**B**) and NAc (**C**,**D**). Data are presented as means ± SEMs. Significant differences between the vehicle group and other groups are indicated by * *p* < 0.05.

## Data Availability

Data is contained within the article.

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
