# Peer review of "Mepirapim, a Novel Synthetic Cannabinoid, Induces Addiction-Related Behaviors through Neurochemical Maladaptation in the Brain of Rodents"

_pharmaceuticals, 2022, doi:10.3390/ph15060710_

Round 1

Reviewer 1 Report

In this manuscript, Hur et al. explore the addictive effects of Mepirapim, a synthetic cannabinoid that has recently been abused for recreational purposes. This is a very interesting and highly relevant study, the manuscript is well written (and already highly polished) and I have only a few comments to be addressed before the manuscript is accepted for publishing.

1. The number of animals (rats and mice) used in each experimental group should be clearly stated I material and methods section as well as in the figures.

2. Figures – in each figure (e.g. bar graphs) individual animal data should also be shown as this is essential for the reader to judge the data spread and quality.

3. 3mg/kg treatment induced hypomotility and hypothermia but not CPP – how do you reconcile these?

Author Response

Reviwer 1

Comments and Suggestions for Authors

In this manuscript, Hur et al. explore the addictive effects of Mepirapim, a synthetic cannabinoid that has recently been abused for recreational purposes. This is a very interesting and highly relevant study, the manuscript is well written (and already highly polished) and I have only a few comments to be addressed before the manuscript is accepted for publishing.

Point 1. The number of animals (rats and mice) used in each experimental group should be clearly stated I material and methods section as well as in the figures.

Response 1. Thank you for pointing out a very important issue. In the Materials and Methods section, we have presented the exact number of animals used in each experiment. In the figure, we have presented the exact number of animals used in each experiment as each point.

Point 2. Figures – in each figure (e.g. bar graphs) individual animal data should also be shown as this is essential for the reader to judge the data spread and quality.

Response 2. Thank you for pointing out a very important issue. In the figure, we have presented the exact number of animals used in each experiment as each point.

Point 3. 3mg/kg treatment induced hypomotility and hypothermia but not CPP – how do you reconcile these?

Response 3. Thank you for pointing out a very critical issue. We added the following comments to the Discussion section. In the CPP test, we found that only low doses of Mepirapim (0.3 and 1 mg·kg−1) induced CPP whereas higher doses (3 mg·kg−1) rather induced conditioned place aversion. This shift from preference to aversion according to the dose increase was also confirmed in other SCBs [1], and it is considered that the dose increase induces the stronger aversive effects than the rewarding effects of the drug. As experimental results supporting this hypothesis, Mepirapim 3 mg/kg induced negative physical symptoms, including hypomotility and hypothermia, that were not induced by lower doses of Mepirapim. Taken together, these results indicate that Mepirapim induces a rewarding effects at low doses and, conversely, aversive effcets at high doses.

  1. Cha, H.J.; Lee, K.-W.; Song, M.-J.; Hyeon, Y.-J.; Hwang, J.-Y.; Jang, C.-G.; Ahn, J.-I.; Jeon, S.-H.; Kim, H.-U.; Kim, Y.-H. Dependence potential of the synthetic cannabinoids JWH-073, JWH-081, and JWH-210: in vivo and in vitro approaches. Biomolecules & therapeutics 2014, 22, 363.

Reviewer 2 Report

The manuscript highlights how research on natural or synthetic cannabinoids have a strong therapeutic interest.

the manuscript is interesting.

Do the authors think the microbiota can influence the effects of mepiramin?

the authors could add a graphical abstract, so the results are immediate.

Could the authors better explain its potential clinical use in the conclusions?

Author Response

Reviwer 2

Comments and Suggestions for Authors

The manuscript highlights how research on natural or synthetic cannabinoids have a strong therapeutic interest.

Point 1. Do the authors think the microbiota can influence the effects of mepiramin?

Response 1. Thanks for the interesting question. I have found many interesting studies that the microbiome metabolizes psychotropic drugs, thereby changing the pharmacokinetic and pharmacodynamic properties of drug [2,3]. Unfortunately, however, since Mepirapim is a novel drug, there are still few studies on factors, including microbiota, that may influence effects of Mepirapim. Therefore, I would consider the influence of Microbiota on the pharmacological effects of Mepirapim as an interesting follow-up study.

Point 2. The authors could add a graphical abstract, so the results are immediate.

Response 2. Thanks for the kind advice. We have presented a graphical abstract outlining the hypothetical mechanism of Mepirapim addiction.

Point 3. Could the authors better explain its potential clinical use in the conclusions?

Response 3. Thank you for pointing out a very important issue. We added the following comments to the Conclusion. Furthermore, the results of this study should be widely publicized so that substance abusers can recognize for themselves that novel psychotropic drugs have dangerous pharmacological effects, including addictive properties.

  1. Walsh, J.; Griffin, B.T.; Clarke, G.; Hyland, N.P. Drug–gut microbiota interactions: implications for neuropharmacology. British journal of pharmacology 2018, 175, 4415-4429.
  2. Cussotto, S.; Clarke, G.; Dinan, T.G.; Cryan, J.F. Psychotropic Drugs and the Microbiome. Microbes and the Mind 2021, 32, 113-133.